# An Overview of Recent Insights into the Response of TLR to SARS-CoV-2 Infection and the Potential of TLR Agonists as SARS-CoV-2 Vaccine Adjuvants

**DOI:** 10.3390/v13112302

**Published:** 2021-11-18

**Authors:** Mohammad Enamul Hoque Kayesh, Michinori Kohara, Kyoko Tsukiyama-Kohara

**Affiliations:** 1Transboundary Animal Diseases Centre, Joint Faculty of Veterinary Medicine, Kagoshima University, Kagoshima 890-0065, Japan; mehkayesh@pstu.ac.bd; 2Department of Microbiology and Public Health, Faculty of Animal Science and Veterinary Medicine, Patuakhali Science and Technology University, Barishal 8210, Bangladesh; 3Department of Microbiology and Cell Biology, Tokyo Metropolitan Institute of Medical Science, Tokyo 156-8506, Japan; kohara-mc@igakuken.or.jp

**Keywords:** SARS-CoV-2, TLRs, TLR agonist, vaccine, adjuvants

## Abstract

The emergence of severe acute respiratory syndrome coronavirus 2 (SARS-CoV-2) has led to coronavirus disease (COVID-19), a global health pandemic causing millions of deaths worldwide. However, the immunopathogenesis of COVID-19, particularly the interaction between SARS-CoV-2 and host innate immunity, remains unclear. The innate immune system acts as the first line of host defense, which is critical for the initial detection of invading pathogens and the activation and shaping of adaptive immunity. Toll-like receptors (TLRs) are key sensors of innate immunity that recognize pathogen-associated molecular patterns and activate downstream signaling for pro-inflammatory cytokine and chemokine production. However, TLRs may also act as a double-edged sword, and dysregulated TLR responses may enhance immune-mediated pathology, instead of providing protection. Therefore, a proper understanding of the interaction between TLRs and SARS-CoV-2 is of great importance for devising therapeutic and preventive strategies. The use of TLR agonists as vaccine adjuvants for human disease is a promising approach that could be applied in the investigation of COVID-19 vaccines. In this review, we discuss the recent progress in our understanding of host innate immune responses in SARS-CoV-2 infection, with particular focus on TLR response. In addition, we discuss the use of TLR agonists as vaccine adjuvants in enhancing the efficacy of COVID-19 vaccine.

## 1. Introduction

Coronaviruses (CoVs) are agents of emerging and re-emerging infection that pose a significant challenge to human health [1]. CoVs have a wide host range, including a variety of avian and mammalian species [2,3]. CoVs are members of the family *Coronaviridae*, which has four genera, including *Alphacoronavirus*, *Betacoronavirus*, *Gammacoronavirus*, and *Deltacoronavirus* [4]. The first human CoV, B814, was reported in 1965 [5]. To date, seven human CoVs (HCoVs) have been identified, including two alpha-CoVs (HCoV-229E and HCoV-NL63) and five beta-CoVs (HCoV-OC43, HCoV-HKU1, severe acute respiratory syndrome (SARS)-CoV, Middle East respiratory syndrome (MERS)-CoV, and SARS-CoV-2) [4]. However, SARS-CoV-2 is one of the most pathogenic and highly infectious zoonotic CoVs. SARS-CoV-2 contains a positive-sense single-stranded RNA genome, with a size of 29.9 kb [6]. It contains 4 structural proteins, including surface (S), envelope (E), membrane (M), and nucleocapsid (N) [7]; 16 non-structural proteins (NSP1–16); and nine accessory proteins [8]. The ongoing coronavirus disease (COVID-19) pandemic was caused by SARS-CoV-2 [9], which is the third zoonotic CoV infecting humans in a short span of only two decades, after the emergence of SARS-CoV in 2002 [10] and MERS-CoV in 2012 [11]. Since the emergence of SARS-CoV-2 in Wuhan, China in December 2019, it has rapidly spread worldwide, and as of 30 September 2021, there have been 233,136,147 confirmed cases of COVID-19, including 4,771,408 deaths globally [12]. The COVID-19 epidemic was declared a global pandemic on 11 March 2020 by the World Health Organization and still continues [13]. The immunopathogenesis of SARS-CoV-2 remains unclear. In addition, several new variants of the virus have also evolved, which may threaten the success of the currently available SARS-CoV-2 vaccines [14]. Thus, there is a need for further investigation of the host immune response to SARS-CoV-2 infection, including newly emerging variants, as well as the efficacy of vaccines against them.

The innate immune system is a key component of host immunity and acts as the first line of defense against invading pathogens, including viruses [15]. The innate immune response is elicited upon detection of conserved structures on microbes, which are known as pathogen-associated molecular patterns (PAMPs) [16]. The key innate immune sensing receptors are germ line-encoded pattern-recognition receptors (PRRs), which mediate the initial sensing of infection by recognition of PAMPs, upon microbial invasion of the host [17,18]. PRRs recognize molecules released by damaged cells, which are known as damage-associated molecular patterns [19]. PRRs belong to different families, including Toll-like receptors (TLRs), retinoic acid-inducible gene I (RIG)-like receptors, nucleotide-binding oligomerization domain-containing protein-like receptors, C-type lectin receptors, and DNA-sensing receptors [16,17]. TLRs are the key components of innate immunity that are evolutionarily conserved and play an important role in early host defense [15,17], by recognizing PAMPs of invading microorganisms, initiating innate immune responses, and shaping adaptive immunity [20,21,22]. TLRs are encoded by a large gene family that includes 10 and 12 members in humans (TLR1–TLR10) and mice (TLR1–TLR9 and TLR11–TLR13), respectively [23]. TLRs can be localized either on the cell surface, such as TLRs 1, 2, 4, 5, 6, and 10, or on the endoplasmic reticulum, such as TLRs 3, 7, 8, and 9 [24,25]. TLR response ultimately may lead to the induction of interferons (IFNs), cytokines, and chemokines by several distinct signaling pathways, thereby limiting infection and promoting adaptive immune responses [17,26]. However, TLR activation may act as a double-edged sword, and its dysregulated response may lead to immune-mediated pathology, instead of protection [27,28,29], which has also been observed in cases of SARS-CoV-2 infection [30]. Therefore, a clear understanding of the role of TLRs in SARS-CoV-2 infection is critical for learning about the immunopathogenesis involved and for the development of therapeutic and preventive strategies against the disease. The focus of this article is to provide an overview of our recent understanding of TLR response to SARS-CoV-2 infection. In addition, the potential of TLR agonists as adjuvants for COVID-19 vaccines has also been discussed herein.

## 2. TLR Response to SARS-CoV-2 Infection

The innate immune response is a key issue to act as the first line of defense against many viral infections [31]. However, the innate immune response does not always work for the host defense but also induces pathogenesis [32]. TLRs constitute the most important family of PRRs. TLR signaling is involved in the regulation of both pro- and anti-inflammatory cytokines secretion working for early innate immune responses and adaptive immunity [33]. TLR activation can act as a double-edged sword, which may activate immune-mediated pathogenesis instead of inducing an immune response which works for defense against pathogens [27,28,34].

The host immune response to SARS-CoV-2 infection is critical in determining the severity of COVID-19 [35]. It is assumed that cytokine storms, which are the consequence of hyperinflammation driven by innate immunity, play an important role in the pathogenesis of severe COVID-19 [36]; however, the underlying mechanisms of the altered pathological inflammation in COVID-19 remain largely unknown. Nevertheless, previous studies on related CoVs, including SARS-CoV and MERS-CoV [37,38,39,40], have enhanced our understanding of SARS-CoV-2 infection. Moreover, in silico studies have also promoted the understanding of SARS-CoV-2 interactions in the host. It has been indicated that cell surface TLRs, mainly TLR4, are most likely to be involved in sensing molecular patterns, including SARS-CoV-2 S protein, to induce inflammatory responses [41,42]. It has been reported that the SARS-CoV-2 S protein S1 subunit can induce pro-inflammatory cytokines through TLR4 signaling in murine and human macrophages, and inhibition of TLR4 by using its antagonist attenuates pro-inflammatory cytokine induction [43], suggesting that TLR4 is a therapeutic target for controlling COVID-19 severity caused by TLR4-mediated hyperinflammation. Another in vitro study also demonstrated the sensing of SARS-CoV-2 S protein by TLR4 and subsequent induction of interleukin (IL)-1B [44]. It has also been indicated that SARS-CoV-2 S protein can bind to bacterial lipopolysaccharide, a ligand for TLR4 activation [45]. SARS-CoV-2-mediated TLR4 and TLR7 upregulation upon periodic thermomechanical modulation has been observed in adipose-derived mesenchymal stromal cells [46]. The expression of interleukin-1 receptor-associated kinase M (IRAK-M), a negative regulator of TLR signaling [47], is suppressed by SARS-CoV-2 S protein in macrophages [48], which may promote pro-inflammatory cytokine production.

It is considered that pro-inflammatory cytokines greatly contribute to the pathogenesis of COVID-19 and its severity. Zheng et al. reported the sensing of SARS-CoV-2 E protein by TLR2, which results in the hyperexpression of pro-inflammatory cytokines (Figure 1) that may contribute to disease severity [30], suggesting that TLR2-mediated inflammation plays a pathogenic role in SARS-CoV-2 infection. TLR1, TLR4, TLR5, TLR8, and TLR9 expression levels were also significantly elevated in severe and critical COVID-19 patients; however, TLR3 expression was not correlated with the development of COVID-19, and an increased expression of TLR7 was observed only in patients with moderate COVID-19 [30]. In a preprint study, another group reported TLR2 activation by SARS-CoV-2 S protein, but not the E protein, and subsequent production of cytokines and chemokines (Figure 1) in human and mouse macrophages [49]. Although this study supports the sensing of SARS-CoV-2 by TLR2, it differs in terms of the sensing protein reported in a previous study [30], which requires further investigation [50].

Another study reported that alveolar macrophages activated by SARS-CoV-2 through TLR signaling may produce IL-1, which further stimulates mast cells to produce IL-6 [51]. In addition, an association between mast cell/eosinophil activation and COVID-19 inflammation was reported [52]. A significantly elevated expression of IL-6 and tumor necrosis factor-α (TNF-α) was found to be associated with TLR expression in obese individuals but not in the controls [53]. TLR/myeloid differentiation factor 88 (MYD88) signaling, which is upregulated in obese individuals, may contribute to the excessive inflammatory response observed in severe infection with SARS-CoV-2 [54].

It is also hypothesized that desensitization of TLR7 signaling may occur due to chronic stimulation of TLR7 by intrinsic substrates in obese and elderly people. This may support the virus to replicate more easily, which upon re-sensitization of TLR7 signaling due to severe viral infection could result in an overwhelming TLR7 response that may promote the development of severe COVID-19 [55]. Several studies have shown that the TLR7 response is critical for favorable outcome of COVID-19 [56,57], highlighting the clinical importance of the innate immune system in SARS-CoV-2 infection. However, in an in vitro study, it has been shown that SARS-CoV-2 induces a TLR7/8-dependent type I and III IFN response in peripheral blood mononuclear cells, which could be protective or may contribute to the cytokine storm observed in COVID-19 [58] and requires future investigation. Although there is a need for further investigation in this regard, IFN-α2b treatment has been shown to reduce viral replication, as well as IL-6 and CRP levels [59]. There was an upregulation in the levels of TLR4-mediated inflammatory signaling molecules in peripheral blood mononuclear cells from COVID-19 patients (Figure 1), as compared to those in the healthy controls, which may suggest an involvement of TLR4 signaling in the induction of pathological inflammation during COVID-19, suggesting that targeting TLR4-mediated inflammation may serve as a new therapeutic strategy [60]. Therefore, TLR2 and TLR4 signaling might be involved in the induction of pro-inflammatory mediators. Infrared light therapy has been shown to decrease TLR4-dependent induction of IL-6, IL-8, TNF-α, INF-α, and INF-β in COVID-19 hyperinflammation [61]. It was observed that a child with hepatitis after SARS-CoV-2 infection had a polymorphism, Gln11Leu (rs179008), in TLR7 [62], which could impair an efficient initial immune response. Elevated cytokine levels have been linked to SARS-CoV-2 infection in adults (median age, 51 years) [63]. Another study observed elevated levels of IL-6 and TNF-α in children with COVID-19, as compared to those in controls; however, the increased levels of these cytokines lacked any correlation with disease severity [64]. It has also been reported that human biological sex, including sex steroids, sex chromosomes, and genomic and epigenetic differences between two sexes, might play a significant role in heterogeneous COVID-19 outcomes by impacting host immune response to SARS-CoV-2 infection [65]. It has been reported that sensing of viral RNA by TLR7 is sex-biased, where TLR7 escapes X chromosome inactivation, resulting in greater expression in female immune cells [66]. However, TLR response in SARS-CoV-2 infection based on sex differences largely remains to be investigated. Aging impacts both innate and adaptive arms of the immune system, and TLR expression and function may decline with increased age, which may affect controlling of viral infections [67,68]. Aging is linked to high morbidity and mortality in various infections [69], which is also applicable for SARS-CoV-2 infection [70,71]. However, the exact role of aging in TLR response against SARS-CoV-2 infection remains to be investigated. A multi-omics study of immunological responses in COVID-19 patients also revealed increased cytokines and chemokines levels in severe COVID-19 patients [72], which is also consistent with the findings of a previous study [73].

IFN signaling cascade is crucial for controlling viral infection. It has been demonstrated that hypoxia-inducible factor-1α (HIF-1α) is a direct transcriptional suppressor of interferon regulatory factors (IRFs), the transcriptional activators of type-I IFN, and hypoxia suppresses type-I IFN but not NF-κB-dependent pro-inflammatory cytokine production [74], which may at least partly explain COVID-19 pathogenesis. Increased expression of HIF-1α mRNA has also been reported in myeloid blood cells from critically ill COVID-19 patients, along with the expression of other genes, TLR2 and TLR4 [75], which could be involved in SARS-CoV-2 sensing. A predominance of cells co-expressing HIF-1α and TLR2 has also been reported [75]. Another study revealed that peripheral blood immune cells from severe COVID-19 patients have diminished type-I IFN response but enhanced pro-inflammatory IL-6 and TNF-α responses [76].

In an in vitro study, it was shown that there was no inhibitory effect of famotidine, a histamine-2 receptor antagonist, on SARS-CoV-2 proteases [77]; however, although famotidine has been found to reduce the risk of intubation and death in COVID-19 patients [78,79], the mechanism involved remains unknown. Recently, it has been reported that famotidine inhibits TLR3-dependent signaling processes that culminate in the activation of IRF3 and the NF-κB pathway in SARS-CoV-2 infection [80]. In response to SARS-CoV-2 infection in human induced pluripotent stem cell (iPSC)-derived lung organoids, most of the key genes associated with innate immunity, cytokine/chemokines, and inflammasomes, including STAT1/2, IRF7, CCL5, CXCL10, TNF-α, IL-6, IL-8, and IFN were upregulated. In the case of human iPSC-derived neuronal cells, in response to SARS-CoV-2 infection, TLR3, TLR7, OAS2, complement system, and apoptotic genes were found to be activated [81]. These findings may further enhance our understanding of COVID-19 pathogenesis. Lactoferrin, a naturally occurring non-toxic glycoprotein of the transferrin family, which is synthesized by exocrine glands and neutrophils, has been shown to partially inhibit SARS-CoV-2 replication in Caco-2 intestinal epithelial cells and also induce the expression of TLR3, TLR7, IFNA1, IFNB1, IRF3, IRF7, and MAVS genes [82], which could be linked to the suppression of viral replication.

Type I and III IFNs are key antiviral mediators against SARS-CoV-2 infection. An in-depth analysis of the transcriptional response to SARS-CoV-2 revealed suppression of type I and III IFN responses, in addition to induction of chemokine and pro-inflammatory cytokine gene expression [83]. One preprint study demonstrated the restriction of SARS-CoV-2 spread by a local IFN-I/III response produced by pDCs; in addition, pDCs response was found to be correlated with the severity of the disease, and the response was impaired in severe COVID-19 patients [84]. Bastard et al. reported that neutralizing auto-antibodies (Abs) against type I IFNs associated with severe COVID-19, of which male patients had the vast majority (95/101, 94%), is suggestive of gender linkage to pathogenic gene encoding auto-Abs [85]. SARS-CoV-2 also possesses the strategies to escape the innate immune response by encoding a wide range of viral structural and nonstructural protein (nsp), affecting the IFN signaling pathway and impairing the IFN-mediated antiviral responses [86,87]. It has been shown that SARS-CoV-2 ORF9b negatively regulates antiviral immunity by suppressing several components of innate immune signaling, including RIG-I, MDA-5, MAVS, TBK1, and IKKε, thereby facilitating viral replication [88]. It has been reported that SARS-CoV-2 nsp6 and nsp13 inhibit IRF-3 phosphorylation by binding TANK binding kinase 1 (TBK1), resulting in a reduced production of IFN-β, while the ORF6 prevents IRF-3 nuclear translocation by binding importin Karyopherin α2 [87,89]. Another study found that SARS-CoV-2 ORF6, ORF8, and nucleocapsid proteins can inhibit type I IFN signaling pathway [90]. It has been demonstrated that SARS-CoV-2 M protein can suppress the expression of IFN-β expression through ubiquitin-mediated degradation of TBK1 [91]. For further details on the immune evasion strategies of SARS-CoV-2, please see the recently published review [92]. However, future investigations are required for clear understanding of the inhibition of TLR signaling by SARS-CoV-2. Therefore, further investigations are required for a clear understanding of the innate immune response, including TLR response, which may open up a new window for therapeutic and preventive strategies against SARS-CoV-2 infection.

## 3. TLR Agonists as COVID-19 Vaccine Adjuvants

There is a growing interest in the use of TLR agonists as immunomodulators for the treatment of inflammation, cancer, infection, allergy, and autoimmunity. The use of TLR agonists to modify immunotherapeutic effects also appears promising [93,94,95,96,97,98,99]. Moreover, TLRs, which are considered as important triggering molecules of trained immunity, trigger a long-term boosting of innate immune responses [100]. To combat the COVID-19 pandemic, there is a critical need for the development of efficient and safe vaccines. Vaccine adjuvants are important for enhancing the immune response against corresponding pathogens, and selecting an appropriate adjuvant is important for vaccine efficacy [101]. It has been reported that second-generation adjuvants that interact with TLRs, such as TLR ligand adjuvants, are superior to first-generation adjuvants, such as Al(OH)_3_. TLR agonist adjuvants may induce dendritic cell maturation, which is lacking in first-generation adjuvants, such as Al(OH)_3_ [102]. TLR agonists are capable of stimulating innate immune responses, which also trigger adaptive immune responses, thereby improving vaccine efficacy.

Subunit vaccines, for which adjuvants are required to enhance the magnitude and durability of immune responses, are the safest and most widely used vaccine platforms that are suitable against a multitude of infectious diseases. In a study of SARS-CoV-2 subunit vaccines, the combined use of TLR1/2 and TLR3 agonist (L-pampo) was found to be a potent adjuvant; the SARS-CoV-2 antigens, along with L-pampo, induced strong humoral and cellular immune responses, with a substantial decrease in viral load, in a ferret model [103]. In another study, it was shown that the TLR2/6 agonist INNA-051 significantly reduced viral RNA levels in throat swabs (96% reduction) and nasal wash (93% reduction) [104], which requires further investigation and may support the clinical development of a therapy based on the prophylactic use of TLR2/6 agonist. The first plant polymer-based TLR4 agonist, inulin acetate, which is synthesized from plant polysaccharide inulin [105], has been reported to induce strong systemic and mucosal immunity [106] and may be useful in the development of a COVID-19 vaccine. CpG ODN, Poly I:C, and resiquimod (R848), which are agonists of TLR9, TLR3, and TLR7/8, respectively, were evaluated in candidate vaccines against SARS-CoV [107], which also supports an investigation into SARS-CoV-2 vaccine.

In a previous study, CD8+ T cells were found to be augmented to varying degrees by CpG ODN, PolyI:C, and R848 [107]; it has also been reported that CD8+ T cell responses might play significant role in preventing SARS-CoV-2 infection [108,109]. However, alum lacks the ability to stimulate CD4+ and CD8+ T cell responses, which has been shown to coordinate with the antibody responses toward protective immunity against SARS-CoV-2 [110]. Recently, the efficacy of a COVID-19 subunit vaccine in promoting protective immunity against SARS-CoV-2 was investigated in rhesus macaques [111]. The vaccine contained the SARS-CoV-2 S protein receptor-binding domain displayed on an I53-50 protein nanoparticle scaffold with five different adjuvants, including a squalene-in-water emulsion (Essai O/W 1849101), an α-tocopherol-containing oil-in-water emulsion (AS03), a TLR7 agonist adsorbed to alum (AS37), a TLR9 agonist formulated in alum (CpG1018-alum), and alum. Notably, variations in the neutralizing antibody production were observed due to the adjuvant differences [111], which indicates that the selection of an adjuvant is critical for vaccine efficacy. In a phase I, randomized, double-blind, placebo-controlled trial, the SCB-2019 vaccine, comprising of S-trimer protein formulated with either AS03 or CpG/Alum adjuvants, induced strong humoral and cellular immune responses against SARS-CoV-2, with high viral neutralizing activity [112]. Both the adjuvanted vaccine formulations were well tolerated and supported further clinical development [112]. It has been suggested that TLR agonists, including imiquimod, an immune stimulator of TLR7, could serve as an effective therapeutic approach in the early stages of COVID-19 [113] and could be more favorable. However, so far, no phase II and III clinical trial results of TLR agonists for COVID-19 vaccines have been published. The TLR agonists currently under development for COVID-19 vaccines are listed in Table 1.

## 4. Conclusions

Current evidence suggests that the hyperinflammation that results from a dysregulated host innate immune response has a negative effect on the COVID-19 outcome. The innate immune response appears to be a double-edged sword in SARS-CoV-2 infection. Its dysregulated signaling may lead to the production of detrimental pro-inflammatory cytokines and chemokines that cause severe disease, instead of providing protection [30]. In this review, we discussed the important role of TLRs in shaping the innate response in SARS-CoV-2 infection. From the currently available data, it is assumed that TLRs, mainly TLR2 and TLR4, may play a pathogenic role by inducing hyperinflammation, and thus, may lead to severe COVID-19. The use of TLR antagonists targeting TLR2 and TLR4 might exert a beneficial effect, by attenuating the deleterious hyperinflammatory response triggered by TLR2 and TLR4 upon sensing SARS-CoV-2 S or E protein; thus, this approach warrants further investigation. In addition, a timely optimum TLR response, mainly mediated by TLR3 and TLR7, could play a protective role against SARS-CoV-2 infection. Application of adjuvant with RBD-NP can enhance neutralizing antibody production and CD4 T cell responses through TLRs. It is understood that optimal IFN production and controlled inflammation are necessary for reducing COVID-19 pathogenesis caused by excessive cytokine production, which could be achieved through the regulation of the TLR-mediated response [114,115]. Therefore, it is critical to obtain a clear understanding of TLR interactions in SARS-CoV-2 infection, which may provide a new basis for the development of therapeutic and preventive approaches to fight the disease, including a TLR agonist-adjuvanted SARS-CoV-2 vaccine.

## Figures and Tables

**Figure 1 viruses-13-02302-f001:**
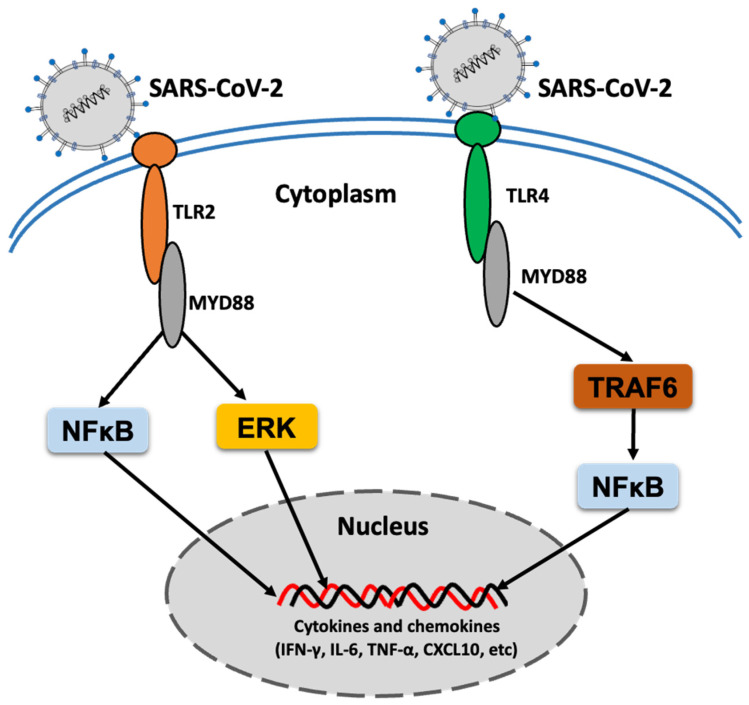
A simplified signaling of TLR2 and TLR4 response to SARS-CoV-2 infection. Induction of inflammatory cytokines and chemokines through TLR2 and TLR4 signaling pathway in response to SARS-CoV-2 infection has been indicated.

**Table 1 viruses-13-02302-t001:** TLR agonists as vaccine adjuvants for enhancing COVID-19 vaccine efficacy.

Vaccine	Sponsor	TLR Agonist Adjuvant	Target TLR	Clinical Phase of Development	Effects on Host Immunity	Clinical Trials. Gov Identifier/Reference
IMP CoVac-1 (SARS-CoV-2-derived multi-peptide vaccine)	University Hospital Tuebingen	TLR1/2 ligand XS15	TLR1/2	Phase I	Results not published yet	NCT04546841
VXA-CoV2-1	Vaxart	dsRNA	TLR3	Phase I	Results not published yet	NCT04563702
VXA-CoV2-1.1-S	Vaxart	dsRNA	TLR3	Phase II	Results not published yet	NCT05067933
SCB-2019 vaccine	Clover Biopharmaceuticals AUS Pty Ltd.	CpG 1018 plus alum	TLR9	Phase I	Well-tolerated in healthy volunteers; elicited T-helper-1-biased CD4+ T-cell responses	[112]
SCB-2019 vaccine	Zhejiang Clover Biopharmaceuticals Inc.	CpG 1018 and alhydrogel	TLR9	Phase II	Results not published yet	NCT04954131
SCB-2019 vaccine	Clover Biopharmaceuticals AUS Pty Ltd.	CpG 1018 plus alum	TLR9	Phase III	Results not published yet	NCT05012787

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
