# Peer review of "An Overview of Recent Insights into the Response of TLR to SARS-CoV-2 Infection and the Potential of TLR Agonists as SARS-CoV-2 Vaccine Adjuvants"

_viruses, 2021, doi:10.3390/v13112302_

Round 1

Reviewer 1 Report

The authors provide a review of the toll like receptor response to SARS-CoV-2 virus infection and discuss the potential of TLR agonists as vaccine adjuvants. While the review of the innate immune response involving TLR's is comprehensive and potentially valuable the information is provided in one very long paragraph that detracts from comprehension. It is this reviewers opinion that section 2 needs to be rewritten extensively, including multiple paragraphs, to more clearly articulate how the TLRs respond to infection. For example, start with an overview of the literature regarding the TLR response, then discuss how certain responses may lead to increased pathology and end in potential therapeutic strategies. Consider also adding information on how these responses may differ based on age or sex.

Section 3 needs to be similarly rewritten to break up the information into manageable snippets with a logical flow. This section would also benefit  from an expanded discussion the results from referenced studies. For example, the use of CpG ODN, Poly I:C and R848 were mentioned in the contect of SARS-CoV but it is not clear from the discussion that the CD8 data is from the same study. Similarly the discussion of the subunit vaccines with 5 different adjuvants is insufficiently detailed.

The conclusions would benefit from a discussion of the adjuvants that might be best applied to SARS-CoV-2.

Author Response

The authors provide a review of the toll like receptor response to SARS-CoV-2 virus infection and discuss the potential of TLR agonists as vaccine adjuvants. While the review of the innate immune response involving TLR's is comprehensive and potentially valuable the information is provided in one very long paragraph that detracts from comprehension. It is this reviewers opinion that section 2 needs to be rewritten extensively, including multiple paragraphs, to more clearly articulate how the TLRs respond to infection. For example, start with an overview of the literature regarding the TLR response, then discuss how certain responses may lead to increased pathology and end in potential therapeutic strategies. Consider also adding information on how these responses may differ based on age or sex.

Response: According to the reviewer comment, we have broken the single paragraph into several paragraphs. Accordingly, we also updated the text including the information as commented by the reviewer (page 2, line 90-97; line 99-100; page 3, line 122-123, line 149-153; page 4, line 167-168, line 176-190).

Section 3 needs to be similarly rewritten to break up the information into manageable snippets with a logical flow. This section would also benefit  from an expanded discussion the results from referenced studies. For example, the use of CpG ODN, Poly I:C and R848 were mentioned in the contect of SARS-CoV but it is not clear from the discussion that the CD8 data is from the same study. Similarly the discussion of the subunit vaccines with 5 different adjuvants is insufficiently detailed.

Response: According to the reviewer comment, we have broken the single paragraph into several paragraphs. In addition, we also updated the text including the suggested information by the reviewer (page 6, line 292-298, line 302).

The conclusions would benefit from a discussion of the adjuvants that might be best applied to SARS-CoV-2.

Response: According to the reviewer comment, we have updated the text (page 7, line 344-346, 380-389).

Reviewer 2 Report

Coronavirus Disease 2019 (COVID-19) is a global pandemic disease which is caused by the severe acute respiratory syndrome coronavirus 2 (SARS-CoV-2). However, the immunopathogenesis of COVID-19, particularly the interaction between SARS-CoV-2 and host innate immunity, remains unclear. Toll-like receptors (TLRs) are key sensors of innate immunity that recognize pathogen-associated molecular patterns and activate downstream signaling for proinflammatory cytokine and chemokine production. The authors in this manuscript summarized the recent progress in their understanding of host innate immune responses in SARS-CoV-2 infection, with particular focus on TLR response. In addition, they discuss the use of TLR agonists as vaccine adjuvants in enhancing the efficacy of COVID-19 vaccine. I believe that this manuscript can provide readers with some useful information. What’s more,this manuscript was clearly written and logically organized,and all the data were systematically carried out. So I would therefore recommend this manuscript for publication in Viruses.

Author Response

Coronavirus Disease 2019 (COVID-19) is a global pandemic disease which is caused by the severe acute respiratory syndrome coronavirus 2 (SARS-CoV-2). However, the immunopathogenesis of COVID-19, particularly the interaction between SARS-CoV-2 and host innate immunity, remains unclear. Toll-like receptors (TLRs) are key sensors of innate immunity that recognize pathogen-associated molecular patterns and activate downstream signaling for proinflammatory cytokine and chemokine production. The authors in this manuscript summarized the recent progress in their understanding of host innate immune responses in SARS-CoV-2 infection, with particular focus on TLR response. In addition, they discuss the use of TLR agonists as vaccine adjuvants in enhancing the efficacy of COVID-19 vaccine. I believe that this manuscript can provide readers with some useful information. What’s more,this manuscript was clearly written and logically organized,and all the data were systematically carried out. So I would therefore recommend this manuscript for publication in Viruses.

Response: We are very grateful to the reviewer for his/her sincere comments and kind consideration.

Reviewer 3 Report

In manuscript Viruses-1437082, the authors present an overview of our recent understanding of TLR response in the context of SARS-CoV-2 infection and discuss the ongoing studies on the potential interest to use TLR agonists as adjuvants for vaccines.

This topic is of great interest in the field, and quite novel. To the best of my knowledge, this was thus far not deeply addressed in review articles focusing on SARS-CoV-2. Nonetheless, a recent review Article by Sartorius et al. covered this question, yet this was an overview for different viruses: https://www.nature.com/articles/s41541-021-00391-8.

The authors highlighted the ‘double-edged sword’ of TLR response in SARS-CoV-2 infection (e.g., as stated in the abstract). While this is an important consideration in the field, this aspect is not clearly defined and addressed in the current review article.

Some specific points for improvement of this Review Article should be addressed by the authors as suggested here:

Specific points.

1- TLR-induced signaling leads to two opposites responses/outcomes in SARS-COV-2 infection: i/ an antiviral response associated the expression of type I and III IFNs and an array ISGs, some of which are already known to directly prevent viral replication and spread and ii/ a pro-inflammatory response, which os now well-defined as deleterious to the host in the context of SARS-CoV-2 infection, greatly contributing to the pathogenesis (similarly observed in the context of infections by various for pathogens). This important dual response of TLR should be better addressed and likely includes as different sections.

2- Along the same line, the authors should better explain the deleterious impact of the pro-inflammatory cytokines, now well-recognized as an important hallmark of COVID-19 severity.

3- In section 2 ‘TLR response to SARS-CoV-2 infection’, some aspects greatly connected to this particular topic should be better addressed, including: i/ some of the numerous mechanisms of viral escape, hereby SARS-CoV-2 inhibits the TLR-induced signal (this can also be shown in a schema), and ii/ important recent publications demonstrating that the importance of the antiviral responses mediated via type I and III IFNs, and mainly produced by the plasmacytoid dendritic cells (pDCs).

4- As opposed to this Section 2, the aspects are logically discussed and more clearly explained in the Section 3.

Minor points:

1- Some unnecessary spaces, and typos should be corrected.

2- Some specific molecules should be better defined, e.g. Famotidine, Lactoferrin, etc..

3- The authors should include the recent publications in high-profile journal demonstrating the detection of auto-antibodies against type I IFNs associated to severe COVID-19.

4- The schema of the virus is unnecessary.

Author Response

In manuscript Viruses-1437082, the authors present an overview of our recent understanding of TLR response in the context of SARS-CoV-2 infection and discuss the ongoing studies on the potential interest to use TLR agonists as adjuvants for vaccines.

This topic is of great interest in the field, and quite novel. To the best of my knowledge, this was thus far not deeply addressed in review articles focusing on SARS-CoV-2. Nonetheless, a recent review Article by Sartorius et al. covered this question, yet this was an overview for different viruses: https://www.nature.com/articles/s41541-021-00391-8.

The authors highlighted the ‘double-edged sword’ of TLR response in SARS-CoV-2 infection (e.g., as stated in the abstract). While this is an important consideration in the field, this aspect is not clearly defined and addressed in the current review article.

Some specific points for improvement of this Review Article should be addressed by the authors as suggested here:

Response: We would like to thank the reviewer for his/her sincere comments and suggestions.

Specific points.

1- TLR-induced signaling leads to two opposites responses/outcomes in SARS-COV-2 infection: i/ an antiviral response associated the expression of type I and III IFNs and an array ISGs, some of which are already known to directly prevent viral replication and spread and ii/ a pro-inflammatory response, which os now well-defined as deleterious to the host in the context of SARS-CoV-2 infection, greatly contributing to the pathogenesis (similarly observed in the context of infections by various for pathogens). This important dual response of TLR should be better addressed and likely includes as different sections.

Response: In line with the reviewer comments, we have addressed the into different sections, and updated the text as per the reviewer comments (page 5, line 199; line 230-239; 242-248; page 6, line 254-256).

2- Along the same line, the authors should better explain the deleterious impact of the pro-inflammatory cytokines, now well-recognized as an important hallmark of COVID-19 severity.

Response: According to the reviewer comments, we have updated the text (page 2, line 99-100; page 3, line 122-123; page 4, line 167-168).

3- In section 2 ‘TLR response to SARS-CoV-2 infection’, some aspects greatly connected to this particular topic should be better addressed, including: i/ some of the numerous mechanisms of viral escape, hereby SARS-CoV-2 inhibits the TLR-induced signal (this can also be shown in a schema),

Response: According to the reviewer comments, we have included the mechanisms of viral escape. In addition, we have referred a recently published review paper (ref 92), which also presented the pictorial presentation of immune evasion, for further details (page 4 line 242-page 6 line 256).

and ii/ important recent publications demonstrating that the importance of the antiviral responses mediated via type I and III IFNs, and mainly produced by the plasmacytoid dendritic cells (pDCs).

Response: According to the reviewer comments, we have included the recent publications highlighting the importance of the antiviral responses mediated via type I and III IFNs  (page 5 line 227, 230-233).

4- As opposed to this Section 2, the aspects are logically discussed and more clearly explained in the Section 3.

Response: We thank the reviewer for his/her sincere comments.

Minor points:

1- Some unnecessary spaces, and typos should be corrected.

Response: We have checked and corrected spaces and typos found.

2- Some specific molecules should be better defined, e.g. Famotidine, Lactoferrin, etc..

Response: According to reviewer comments, we have defined these molecules.

3- The authors should include the recent publications in high-profile journal demonstrating the detection of auto-antibodies against type I IFNs associated to severe COVID-19.

Response: According to reviewer comments, we have included the recent publication of demonstrating auto-antibodies against type I IFNs associated to severe COVID-19 (reference 85).

4- The schema of the virus is unnecessary.

Response: According to the reviewer comments, we have deleted the schema of the virus.

Reviewer 4 Report

This is a fine review article that does an excellent job of citing many and diverse publications in the field of TLRs in SARS-CoV-2 response and vaccine development.

__________________________________________________
__________________________________________________

MAJOR CONCERNS

In the Conclusion:
"The innate immune response, particularly TLR response, appears to be a double-edged sword in SARS-CoV-2 infection, and its dysregulated signaling may lead to the production of detrimental pro-inflammatory cytokines and chemokines that cause severe disease, instead of providing protection."

Have you really ruled out the other players in the innate immune system, including all the other PRRs that you mention in the Introduction? I don't think you do. You do a great job of making a case that TLRs are very important. But you don't talk much about the other factors not also being very important. So I don't think you can write "particularly". I think you need to write something like:

"The innate immune response appears to be a double-edged sword in SARS-CoV-2 infection. Its dysregulated signaling may lead to the production of detrimental pro-inflammatory cytokines and chemokines that cause severe disease, instead of providing protection. In this review we discussed the important role of TLRs in shaping the innate response in in SARS-CoV-2 infection."

__________________________________________________

Section 2 is a single paragraph that runs about 2.5 pages. A lot of references are packed in there, which is good if I am using this review article to identify a bunch of relevant references. But it is hard to learn much here because it all comes as water from a fire hose. Is there any chance you could use multiple paragraphs, perhaps adding an introductory and conclusion sentence to each paragraph in order to focus the reader on the concept the paragraph is devoted to?

Section 3 is also a single paragraph that would be easier to follow if it was broken into several shorter paragraphs, each with its own theme.

__________________________________________________

Moreover, TLRs, which are considered as important triggering molecules of trained immunity [...] could serve as a tool for reducing the severity of SARS-CoV-2 infection [76]."

I don't see how reference [76] supports this statement. Exactly how can TLRs be used as a tool for reducing the severity of SARS-CoV-2 infection? And what exactly is written in [76] to support this claim?

__________________________________________________
__________________________________________________

MINOR POINTS

"Therefore, a proper understanding of the balanced interaction between TLRs and SARS-CoV-2 is"

delete the word "balanced". That adjective implies an exactly equal interaction. You don't need an adjective here, because you have already written what you need to write in the previous sentence.

__________________________________________________

delete all the phrases "In a recent study" or similar uses of the word "recent".
EVERY study on COVID19 is recent. 

__________________________________________________

You might add citations to the systems biology of the innate immune system in COVID. E.g.,

Su Y, Chen D, Yuan D, Lausted C, Choi J, Dai CL, Voillet V, Duvvuri VR, Scherler K, Troisch P, Baloni P, Qin G, Smith B, Kornilov SA, Rostomily C, Xu A, Li J, Dong S, Rothchild A, Zhou J, Murray K, Edmark R, Hong S, Heath JE, Earls J, Zhang R, Xie J, Li S, Roper R, Jones L, Zhou Y, Rowen L, Liu R, Mackay S, O'Mahony DS, Dale CR, Wallick JA, Algren HA, Zager MA; ISB-Swedish COVID19 Biobanking Unit, Wei W, Price ND, Huang S, Subramanian N, Wang K, Magis AT, Hadlock JJ, Hood L, Aderem A, Bluestone JA, Lanier LL, Greenberg PD, Gottardo R, Davis MM, Goldman JD, Heath JR. Multi-Omics Resolves a Sharp Disease-State Shift between Mild and Moderate COVID-19. Cell. 2020 Dec 10;183(6):1479-1495.e20. doi: 10.1016/j.cell.2020.10.037. Epub 2020 Oct 28. PMID: 33171100; PMCID: PMC7598382.

and other similar papers. I believe the above paper (although it doesn't mention TLRs per se) shows activation of monocytes and the downstream signaling molecules (e.g., IL-6 and TNFA) consistent with your hypotheses of (1) TLR activation driving severe COVID, (2) less activation in mild COVID.

__________________________________________________

Could you check to see if any of these 3 citations are warranted for inclusion in your review? They came to my attention via a search for "TLR4 [titl] AND adjuvant [titl] AND review" on PubMed.

Evans JT, Cluff CW, Johnson DA, Lacy MJ, Persing DH, Baldridge JR. Enhancement of antigen-specific immunity via the TLR4 ligands MPL adjuvant and Ribi.529. Expert Rev Vaccines. 2003 Apr;2(2):219-29. doi: 10.1586/14760584.2.2.219. PMID: 12899573.

Alderson MR, McGowan P, Baldridge JR, Probst P. TLR4 agonists as immunomodulatory agents. J Endotoxin Res. 2006;12(5):313-9. doi: 10.1179/096805106X118753. PMID: 17059695.

Baldridge JR, McGowan P, Evans JT, Cluff C, Mossman S, Johnson D, Persing D. Taking a Toll on human disease: Toll-like receptor 4 agonists as vaccine adjuvants and monotherapeutic agents. Expert Opin Biol Ther. 2004 Jul;4(7):1129-38. doi: 10.1517/14712598.4.7.1129. PMID: 15268679.

__________________________________________________

Mild grammar/style issues, e.g.

"pathogens, as well as, for the activation and shaping"

better written as

"pathogens and the activation and shaping"

Maybe you wrote they way you did to avoid having two "ands" near each other. But the two "ands" is a better choice. Even better you could rewrite the entire sentence in a creative way or make it into two sentences. But you don't need to.

Try pasting your text in Google Docs and using Google's grammar checker and/or use another grammar checker such as the on Microsoft WORD. Checking grammar is not absolutely necessary. The paper is OK. But it would come across a bit better with better grammar.

Author Response

MAJOR CONCERNS

In the Conclusion:
"The innate immune response, particularly TLR response, appears to be a double-edged sword in SARS-CoV-2 infection, and its dysregulated signaling may lead to the production of detrimental pro-inflammatory cytokines and chemokines that cause severe disease, instead of providing protection."

Have you really ruled out the other players in the innate immune system, including all the other PRRs that you mention in the Introduction? I don't think you do. You do a great job of making a case that TLRs are very important. But you don't talk much about the other factors not also being very important. So I don't think you can write "particularly". I think you need to write something like:

"The innate immune response appears to be a double-edged sword in SARS-CoV-2 infection. Its dysregulated signaling may lead to the production of detrimental pro-inflammatory cytokines and chemokines that cause severe disease, instead of providing protection. In this review we discussed the important role of TLRs in shaping the innate response in in SARS-CoV-2 infection."

Response: We are very grateful to the reviewer for indicating this important point. Accordingly, we have updated the text including the indicated text by the reviewer (page 7, line 369-373).

__________________________________________________

Section 2 is a single paragraph that runs about 2.5 pages. A lot of references are packed in there, which is good if I am using this review article to identify a bunch of relevant references. But it is hard to learn much here because it all comes as water from a fire hose. Is there any chance you could use multiple paragraphs, perhaps adding an introductory and conclusion sentence to each paragraph in order to focus the reader on the concept the paragraph is devoted to?

Response: In line with the reviewer comments, we have modified the section 2 into multiple paragraphs, including an introductory and conclusion sentence.

Section 3 is also a single paragraph that would be easier to follow if it was broken into several shorter paragraphs, each with its own theme.

Response: In line with the reviewer comments, also we have modified the section 3 into several shorter paragraphs.

__________________________________________________

Moreover, TLRs, which are considered as important triggering molecules of trained immunity [...] could serve as a tool for reducing the severity of SARS-CoV-2 infection [76]."

I don't see how reference [76] supports this statement. Exactly how can TLRs be used as a tool for reducing the severity of SARS-CoV-2 infection? And what exactly is written in [76] to support this claim?

Response: We regret for any deficiency and confusion of the study. Accordingly, to avoid any confusion we have updated the text deleting this statement.

__________________________________________________
__________________________________________________

MINOR POINTS

"Therefore, a proper understanding of the balanced interaction between TLRs and SARS-CoV-2 is"

delete the word "balanced". That adjective implies an exactly equal interaction. You don't need an adjective here, because you have already written what you need to write in the previous sentence.

Response: In line with reviewer comment we have deleted the word “balanced” (line 23).

__________________________________________________

delete all the phrases "In a recent study" or similar uses of the word "recent".
EVERY study on COVID19 is recent.

Response: We are very grateful to the reviewer for indicating this point. Accordingly, we have deleted all the phrases "In a recent study" or similar uses of the word "recent".

__________________________________________________

You might add citations to the systems biology of the innate immune system in COVID. E.g.,

Su Y, Chen D, Yuan D, Lausted C, Choi J, Dai CL, Voillet V, Duvvuri VR, Scherler K, Troisch P, Baloni P, Qin G, Smith B, Kornilov SA, Rostomily C, Xu A, Li J, Dong S, Rothchild A, Zhou J, Murray K, Edmark R, Hong S, Heath JE, Earls J, Zhang R, Xie J, Li S, Roper R, Jones L, Zhou Y, Rowen L, Liu R, Mackay S, O'Mahony DS, Dale CR, Wallick JA, Algren HA, Zager MA; ISB-Swedish COVID19 Biobanking Unit, Wei W, Price ND, Huang S, Subramanian N, Wang K, Magis AT, Hadlock JJ, Hood L, Aderem A, Bluestone JA, Lanier LL, Greenberg PD, Gottardo R, Davis MM, Goldman JD, Heath JR. Multi-Omics Resolves a Sharp Disease-State Shift between Mild and Moderate COVID-19. Cell. 2020 Dec 10;183(6):1479-1495.e20. doi: 10.1016/j.cell.2020.10.037. Epub 2020 Oct 28. PMID: 33171100; PMCID: PMC7598382.

and other similar papers. I believe the above paper (although it doesn't mention TLRs per se) shows activation of monocytes and the downstream signaling molecules (e.g., IL-6 and TNFA) consistent with your hypotheses of (1) TLR activation driving severe COVID, (2) less activation in mild COVID.

Response: In line with reviewer comments, we have updated the text including the suggested paper (page 4, line 189, ref 72).

__________________________________________________

Could you check to see if any of these 3 citations are warranted for inclusion in your review? They came to my attention via a search for "TLR4 [titl] AND adjuvant [titl] AND review" on PubMed.

Evans JT, Cluff CW, Johnson DA, Lacy MJ, Persing DH, Baldridge JR. Enhancement of antigen-specific immunity via the TLR4 ligands MPL adjuvant and Ribi.529. Expert Rev Vaccines. 2003 Apr;2(2):219-29. doi: 10.1586/14760584.2.2.219. PMID: 12899573.

Alderson MR, McGowan P, Baldridge JR, Probst P. TLR4 agonists as immunomodulatory agents. J Endotoxin Res. 2006;12(5):313-9. doi: 10.1179/096805106X118753. PMID: 17059695.

Baldridge JR, McGowan P, Evans JT, Cluff C, Mossman S, Johnson D, Persing D. Taking a Toll on human disease: Toll-like receptor 4 agonists as vaccine adjuvants and monotherapeutic agents. Expert Opin Biol Ther. 2004 Jul;4(7):1129-38. doi: 10.1517/14712598.4.7.1129. PMID: 15268679.

Response: According to the reviewer comment, we have cited theses suggested papers (ref.97-99)(line 263)

__________________________________________________

Mild grammar/style issues, e.g.

"pathogens, as well as, for the activation and shaping"

better written as

"pathogens and the activation and shaping"

Maybe you wrote they way you did to avoid having two "ands" near each other. But the two "ands" is a better choice. Even better you could rewrite the entire sentence in a creative way or make it into two sentences. But you don't need to.

Response: According to the reviewer comment we have written as "pathogens and the activation and shaping".

Try pasting your text in Google Docs and using Google's grammar checker and/or use another grammar checker such as the on Microsoft WORD. Checking grammar is not absolutely necessary. The paper is OK. But it would come across a bit better with better grammar.

Response: Thank you for your suggestion. Accordingly, we have checked the text.

Round 2

Reviewer 1 Report

Edited manuscript appropriately addresses previous concerns.

Reviewer 3 Report

The authors adequately addressed my comments.